# The Effect of Dielectric Barrier Discharge Plasma Gas and Plasma-Activated Water on the Physicochemical Changes in Button Mushrooms (*Agaricus bisporus*)

**DOI:** 10.3390/foods11213504

**Published:** 2022-11-03

**Authors:** Yan Zheng, Yifan Zhu, Yanhong Zheng, Jiajie Hu, Jing Chen, Shanggui Deng

**Affiliations:** 1College of Food Science and Pharmacy, Zhejiang Ocean University, Zhoushan 316022, China; 2Key Laboratory of Health Risk Factors for Seafood of Zhejiang Province, Zhoushan 316022, China

**Keywords:** button mushroom, atmospheric cold plasma, plasma-activated water immersion, polyphenol oxidase, sensory evaluation

## Abstract

Button mushrooms (*Agaricus bisporus*) are highly popular worldwide due to their rich nutritional value and health benefits. However, the rapid water loss rate and browning restrict their economic value. The atmospheric cold plasma (ACP) generated by the plasma equipment used by dielectric barrier discharge preservation technology is widely used for food preservation since it is cost-efficient and environmentally friendly, generating no chemical residues. This study established four treatment groups, namely the direct ACP treatment group (DBD), plasma-activated water immersion group (PAW), pure water immersion group (PW), and control group (control), to explore the effect that ACP preservation technology has on button mushrooms. The results indicated that ACP treatment decreased the pH of pure water from 5.90 ± 0.03 to 5.16 ± 0.03, while significantly increasing the temperature (*p* < 0.05). During the storage period, the browning index (BI) and E value were the lowest in the PAW group, which exhibited the best hardness and sensory properties. Neither the pH nor water activity changed significantly during the storage period in any of the groups. The polyphenol oxidase (PPO) activity in the button mushroom decreased significantly compared with the control after plasma-activated water treatment. In summary, plasma-activated water significantly reduced the BI and E value of button mushrooms, inhibited PPO activity, and yielded the most stable sensory properties for the optimal preservation of button mushrooms.

## 1. Introduction

Button mushrooms (*Agaricus bisporus*) are the most widely cultivated and consumed edible mushrooms in the world [1]. They are rich in amino acids, nucleotides, vitamin B1, vitamin B2, vitamin C, nicotinic acid, and provitamin D while improving human immunity and anti-cancer resistance [2]. The tyrosinase in button mushrooms can significantly lower blood pressure [3]. Due to environmental stress, the harvested button mushrooms need to rely on their own respiration and transpiration to maintain nutritional balance [4]. With the extension of storage time, the consumption of a large amount of water and energy makes it gradually exhibit mass loss, cap opening, stipe elongation, browning and softening, and even rotting. Color is an important symbol of the freshness of mushrooms. Enzymatic browning is the main reason for the color change of button mushrooms during storage. When phenolic substrates are oxidized to quinones under the catalysis of polyphenol oxidase (PPO) and further form black substances, browning will occur [5]. The temperature and pH value are two main factors affecting PPO activity. The pH value determines the binding degree of PPO with substrate and catalysis. The temperature not only affects oxygen solubility but also may cause denaturation and inactivation of enzymes [6]. Therefore, the pH value and temperature of button mushrooms during storage are very important for keeping them fresh.

Currently, the most commonly used preservation methods include low-temperature refrigeration [7], ozone preservation [8], air-conditioning preservation [9], radiation preservation [10], and biological coating film preservation [11]. However, these preservation methods present disadvantages, such as off-flavors, high cost, decreased nutritive value, or chemical residues [12,13]. In recent years, many environmentally friendly techniques have emerged for preserving fruits, vegetables, and fresh meat. Atmospheric cold plasma (ACP) uses the photoelectrons, ions, and active free radicals generated by the medium around the food to cause cell damage when contacting the microbial surface, achieving sterilization [14,15]. It presents advantages, such as a short action time and a low sterilization temperature. ACP quickly inactivates enzymes such as lysozymes, PPO, and POD, significantly reducing fruit and vegetable browning during storage. Research has explored the preservative effect of plasma-activated water in extending the shelf-life of button mushrooms over 7 d of storage at 20 °C [16]. However, minimal studies are available regarding the use of ACP generated via the dielectric barrier discharge of plasma equipment for the preservation of button mushrooms at +4 °C.

In this study, we introduced the ACP preservation technology into the study of button mushroom freshness change during 4 °C storage. Using dielectric barrier discharge plasma equipment to generate ACP to directly treat button mushrooms (DBD) and dielectric barrier discharge plasma equipment to generate plasma-activated water to soak button mushroom (PAW) immersion to examine the physicochemical changes during storage. The results are compared to two additional groups of mushrooms, one treated with pure water (PW) and one of untreated button mushrooms (control). Postharvest preservation provides new insight and a theoretical basis for conserving fresh food.

## 2. Materials and Methods

### 2.1. The Mushroom Samples and Storage Conditions

The button mushrooms, obtained from City Shop in Zhoushan, China, were at the closed cap stage phase with pile diameters of around 5 cm. They were transferred to the laboratory within 30 min and stored at +4 °C until experimental use.

### 2.2. The Plasma Equipment and Plasma-Activated Water Generation

ACP-generating dielectric barrier discharge plasma equipment was used for direct sample treatment and plasma-activated water. The processing conditions were set according to a method delineated by Gavahian et al. [14] with minor modifications. The processing conditions for plasma-activated water generation and dielectric barrier discharge plasma for direct treatment and plasma-activated water generation were the same and included a working gas of air, a treatment time of 20 min, an output voltage of 50 kV, and a room temperature of 25 °C. All experiments were performed at room temperature. Plasma was discharged above the water surface to obtain the plasma-activated water.

### 2.3. Sample Treatments

The mushrooms (400 ± 10 g) selected for the various treatments were free of damage and were similar in maturity and size. The button mushrooms were randomly divided into two treatment groups (DBD and PAW) and two control groups (PW and control). The PAW and PW mushrooms were soaked for 20 min in 1000 mL of plasma-activated water and pure water, respectively, and shaken and wiped to remove the surface water. The samples directly treated with ACP for 20 min at 50 kV were labeled the DBD group, while the untreated mushrooms not subjected to ACP or soaking were considered the control group. The mushroom specimens were immediately assessed after treatment, placed in double-sided perforated polyethylene Ziploc bags (10 pcs per side), and stored at +4 °C for 5 d. Each treatment group contained 30 replicates, of which three were selected from each group every other day to analyze the physicochemical parameters (Figure 1).

### 2.4. The pH and Temperature Determination of Plasma-Activated Water

The pH and temperature of the plasma-activated water were measured after ACP treatment. The pH was monitored using a pH meter (Thermo Fisher, Shanghai, China). A FLUKE-62max thermometer (Fluke, Nanjing, China) was used to measure the plasma-activated water temperature immediately after ACP treatment. The procedure was repeated three times for each sample, and the results were presented as the average value. Before the experiment (0 d), the original pH value, hardness, water activity, and sensory score of each group of samples were measured, and there was no significant difference between the samples of different groups (*p* > 0.05).

### 2.5. Color Evaluation

A Minolta spectrophotometer (CR-400) was employed to ascertain the mushroom cap surface color using CIE (*L* a* b**), a standard system for assessing changes in color that enables color determination in a 3D space. The degree of blue/yellow, green/red, and brightness of a sample is represented by the b-plane, a-plane, and L-axis [16]. Readings were collected at three evenly spaced locations on each mushroom cap to determine the *b** (yellow/blue), *a** (red/green), and *L** (light/dark) values. Each replication involved analyzing three mushroom caps. The following equation was used to determine the browning index (BI), which symbolizes the purity of the brown hue [17]:(1)$$\mathrm{BI}=[100\;(x-0.31)]\;/\;0.172,\;\mathrm{where}\;x=(a^{*}+1.75L^{*})\;/\;(5.645L^{*}+a^{*}-3.012b^{*})$$

∆*E*, which denotes the general changes in the button mushroom color [18], was determined via the following equation:(2)$$∆E=\sqrt{(L_{c}^{*}-L_{t}^{*}){}^{2}+(a_{c}^{*}-a_{t}^{*}){}^{2}+(b_{c}^{*}-b_{t}^{*}){}^{2}}$$
where $L_{c}^{*}$, $a_{c}^{*}$, and $b_{c}^{*}$ were the color values of the fresh mushrooms without immediate postharvest treatment, while $L_{t}^{*}$, $a_{t}^{*}$, and $b_{t}^{*}$ represented the mushroom color values during storage.

### 2.6. Textural Evaluation

The firmness (indicating softening) of the button mushrooms was evaluated according to a penetration test established by Nasiri et al. [19] and expressed as newton (N). The evaluation was performed using a TMS-PRO texture analyzer (FTC, Sterling, VA, USA) equipped with a 5 mm diameter probe at a penetration depth of 7 mm and a speed of 5 mm/s. The penetration test was conducted daily throughout the storage period. The textural analysis was performed three times for each sample, and the results were reported as the average firmness value.

### 2.7. pH Evaluation

Here, 20–25 g of stored mushrooms were cut into small pieces and manually squeezed in gauze to obtain the juice, which was used to determine the pH value. The pH was measured at room temperature (25 ± 1 °C) using a pH meter (Thermo Fisher, Shanghai, China).

### 2.8. Sensory Evaluation

The sensory evaluation was performed via quantitative descriptive analysis (QDA). A panel of ten members with long-term training and experience in food evaluation assessed the button mushroom samples using different processing methods. The evaluation indicators were color, umbrella opening degree, and wilting degree. Each indicator included six points. The total score was calculated via the weighting method. The color, degree of umbrella opening, and wilting degree were 0.4, 0.4, and 0.2, respectively. A sensory score of 2.8 indicated that the button mushrooms had reached the end of their shelf life. The sensory evaluation standards for button mushrooms are listed in Table 1.

### 2.9. Weight Loss and Water Activity Determination

The button mushroom weight loss was determined by weighing the fruiting bodies throughout the storage period. The results were expressed as the weight loss percentage of the initial weight:(3)$$\mathrm{Weight}\;\mathrm{loss}\;(\%)=\frac{W_{i}-W_{t}}{W_{i}}\times 100$$
where *W_i_* is the initial weight and *W_t_* is the weight during storage.

An HD-3A water activity tester (Huake Instrument Co., Ltd., Wuxi, China) was used to measure the water activity. Three samples were randomly selected from each of the PAW, DBD, control, and PW groups and chopped. Then, 5.0 g samples were evenly distributed on the surface of a plastic dish and sealed with the metal cover of the water activity meter, after which the vapor pressure of the samples and surrounding air was balanced in constant temperature conditions. The water vapor pressure of the gas space was used as the sample value to measure its water activity.

### 2.10. PPO Activity Assay

The PPO activity of the untreated mushrooms and those exposed to ACP for 20 min were measured according to a procedure described by Dedeoglu et al. [20] with minor modifications, using catechol as the substrate. After cold plasma treatment, 5 mg of button mushroom frozen powder was homogenized in 100 mL of sodium phosphate buffer (0.05 M, pH7.4). Of the above solution, 0.5 mL was mixed with 2 mL of 50 mM catechol in sodium phosphate buffer (0.05 M, pH7.4) using a cuvette with a 1 cm path length at 37 °C [21]. The absorbance of the mixture was measured at 420 nm for 2 min at 30-s intervals using a UV-vis spectrophotometer (UNICO, Shanghai, China). The remaining enzyme solution was separated and placed in a refrigerator at 4 °C for continuous measurement for 5 d. A total of 1 enzyme unit (U) was defined as a change in absorbance of 0.001 in 1 min. The enzyme activity was calculated as follows using Equation (4):(4)$$\mathrm{PPO}\;\mathrm{activity}\;(U/\min)=\frac{∆A_{420}}{0.001\times t}$$
where $∆A_{420}$ is the change in absorbance of 420 nm, and *t* is the reaction time (min).

### 2.11. Statistical Analysis

The experiments were performed using a completely randomized design. The values from all the experiments were expressed as the mean ± standard deviation (SD). The data were assessed via analysis of variance (ANOVA) using SPSS 17.0 software (IBM, Armonk, NY, USA). Differences at *p* < 0.05 were considered significant.

## 3. Results and Discussion

### 3.1. pH and Temperature Determination of Plasma-Activated Water

The variation in the temperature and pH of the pure water after ACP treatment for 20 min is presented in Table 2. The water activation via ACP increased the average temperature from 18.80 ± 0.08 °C to 20.47 ± 0.40 °C. Using a different type of plasma system in various conditions might account for the temperature increase variation in this study, compared to the research by Gavahian et al. [14], whose results indicated that water activation via arc plasma increased its average temperature from 9 ± 0 °C to 34 ± 2 °C. The pH of the pure water decreased from 5.90 ± 0.03 to 5.16 ± 0.03 after ACP treatment. Compared to the control (6.06), the pH value of plasma-activated water decreased significantly to 3.57 after 25 min of plasma activation (*p* < 0.05), indicating that plasma discharge could lead to water acidification [22]. A similar report by Ma et al. [23] showed that the pH of plasma-activated water declined rapidly during the first 10 min of plasma activation, reaching a steady state of around 3 after 20 min. It is well-known that the chemical interaction between reactive species (RS) during plasma treatment causes acidification due to hydrogen POD, nitric acid, and peroxynitrous acid formation. The generation of new chemical species in plasma-activated water is associated with a decrease in the water pH during nonthermal plasma activation. Electron spin resonance (ESR) has confirmed the presence of •OH, ^1^O2, and •O_2_ species in the water after plasma treatment [24]. Therefore, plasma-activated water contained several RS.

### 3.2. Physicochemical Properties of the Mushrooms

#### 3.2.1. Cap Browning and Appearance Changes

The visual browning, BI, and *∆E* of the mushrooms increased gradually during 5 d of storage at +4 °C, as shown in Figure 2, but it was significantly lower in the plasma-activated water-treated mushrooms (*p* < 0.05). Of all the treatment groups, the PAW group optimally maintained the cap quality during cold storage. The results indicated that the PAW sample displayed the lowest Δ*E* values and the cap BI for both sides (front and back), suggesting that the color of this sample exhibited more similarities to the fresh mushroom sample than the other samples. Contrarily, direct ACP treatment increased the overall color variation in the mushrooms during storage compared to the PW and control samples. At 5 d, the BI of the DBD group was almost five times that of the PAW group, while no significant differences were evident between the PW and control groups (*p* > 0.05). RS can inactivate enzymes by preventing the substrate and coenzyme from binding and subsequent catalysis, mainly due to conformational changes around the active site and chemical modification of the active site residues [25]. It can be inferred that reactive oxygen species (ROS) and reactive nitrogen species (RNS) can be dissolved in water to continuously inactivate PPO, as shown by the PAW group. However, these active substances may directly react with phenolic substances on the mushroom surfaces to form quinones, accelerating the browning process. This explains the extensive browning in the DBD group. Cooper et al. [26] reported that damage to the delicate contaminated surfaces and biological materials resulting from the electric field of direct plasma treatment and certain physical species, such as charged particles, UV rays, and electrons, can be avoided via plasma-activated water treatment with only RS.

#### 3.2.2. The Effect of Plasma Treatment on Hardness and Weight Loss

The hardness of the mushrooms was used to determine their quality during storage. The metabolism of the fruit body continued for 1–2 d after harvest while the original pectin was still synthesized [27]. The original pectin combined with cellulose provided the fruit with a firm, crisp appearance. However, the protein, polysaccharides, and other nutrients were degraded during storage, causing tissue relaxation. Furthermore, transpiration promoted the loss of natural water from the epidermis, causing the surface of the button mushrooms to become wrinkled and decreasing its hardness [28]. A rapid decrease in hardness caused the storage performance to decline. The hardness of the button mushrooms in the different groups varied with storage time, as shown in Figure 3A. The button mushrooms in all the groups softened gradually throughout postharvest storage, with optimal tissue firmness evident in the PAW group. The plasma-activated water treatment inactivated microbes to delay softening, corresponding with the relative conductivity results. On the final storage day, the firmness values of the control, DBD, and PAW groups were 12.19 N, 12.67 N, and 19.69 N, respectively, which were higher than the PW group. Excess water expedited senescence and improved the respiration rate, causing the PW group to exhibit the highest degradation [29]. Furthermore, the results showed that the PAW group was more durable during storage.

As shown in Figure 3B, all the higher weight loss rates throughout storage. A higher weight loss rate of 0.35% was initially evident in the DBD group after 3 d of storage, while those of the control and PAW groups were 0.29% and 0.15%, respectively. The plasma-activated water-treated mushrooms displayed less significant cap browning and substantially (*p* < 0.05) lower weight loss. The PW group exhibited the most significant weight loss (1.03%) after 5 d of storage since the respiration of the mushrooms could be enhanced by soaking them in distilled water, while the ACP-treated mushrooms presented a value of 0.66%. The weight loss of the mushrooms was substantially affected by substance consumption via respiration and the transpiration of water. Dehydration was a vital factor that determined the quality of the mushrooms during postharvest storage. This may be because the plasma-activated water treatment inhibits the enzyme activity related to mitochondrial respiratory metabolism in the fruit body, resulting in a slower metabolism and reduced weight loss, also implying the benefits of plasma-activated water in postharvest storage conditions.

#### 3.2.3. The Effect of Plasma Treatment on pH

The pH of the PAW, DBD, PW, and control groups at 5 d were 6.08, 6.18, 6.11, and 6.14, respectively (Table 2). The PAW group exhibited the lowest pH value (6.08) at the end of storage (5 d). Furthermore, no significant changes were evident in the sample pH values during storage. The lack of a substantial pH decrease in the samples during storage may be due to the brief lack of a substantial pH decrease in the samples during storage may be due to the brief observation period (5 d of storage) and low temperature (4 °C), preventing mushroom degradation. This may be due to the significantly reduced pH levels in the samples during storage. Previous studies indicated that no changes were evident in the pH of control and plasma-treated cherry tomatoes over a 14-d storage period at 20 °C [30]. Xu et al. [16] found that no significant differences were apparent between the pH values of the control, pure water, and plasma-activated water-treated specimens over one week. The pH of plasma-treated shiitake mushrooms also did not significantly change in one week [14]. The negligible pH differences could be ascribed to minor metabolic alterations.

#### 3.2.4. The Sensory Evaluation and Water Activity of the Mushrooms

Since button mushrooms contain phenols, they are prone to enzymatic browning, while the color change is intuitive and obvious. During storage, the color of the mushrooms gradually darkened from an initial bright white to brown. The umbrella opening was another important parameter to characterize the morphological button mushroom quality. Senescence is a process in which the gills gradually expand from an unopened to a fully opened umbrella, ultimately breaking the protective film to expose the mature gills and release the spores [31]. To ensure the release of the next generation of spores, the nutrients in the fruit body will undergo a certain degree of migration or lack thereof. This leads to changes in the physiology and appearance of button mushrooms, significantly reducing their quality and commercial value. As shown in Figure 4A, the sensory scores of the different treatment groups decreased during storage. The sensory score indicated that the DBD group displayed the fastest decline, presenting 2.80 points over 3 d, with the samples reaching the end of their shelf life. However, at the end of the storage period, the decrease in the control group exceeded the DBD group. In addition, the sensory scores of the PAW and PW groups were 3.60 and 3.13, respectively, showing superior quality to the control and DBD groups. Therefore, plasma-activated water-treated mushrooms are likely to be more popular with consumers when applied during actual production.

Microbial growth in food, enzymatic or non-enzymatic changes, oil oxidation, and reaction speeds are all significantly related to water activity (a_w_). Reducing the water activity in food can delay the progression of enzymatic and non-enzymatic browning to reduce nutrient destruction. However, if the water activity is too low, it accelerates the oxidative rancidity of fat. As shown in Figure 4B, the water activity of each component showed a slight upward trend before 3 d and began to decline after 3 d. However, except for the PW group, the differences between the groups were not obvious during the storage period, with the values ranging between 0.8 and 0.99, indicating high humidity. The results showed that ACP did not significantly affect water activity, regardless of direct or indirect treatment.

#### 3.2.5. PPO Activity Assay

Enzymes like PPO and POD are involved in enzymatic browning reactions and the subsequent loss in nutritional quality. Figure 5 shows the effect of plasma-activated water treatment on the PPO activity during catechol oxidation after 5 d of storage. At 0 d, the PPO activity was measured immediately after plasma-activated water treatment, showing a significant difference (*p* < 0.05) between the treatment and control groups. The activity of the control group was 1788 U/min, while the treatment group was 449 U/min, which was less than 30% of the control group. During the first 2 d of storage, the PPO activity in the control group increased slightly, followed by a gradual decline. The PPO activity remained at 1880 U/min by the last day of storage. Contrarily, the PPO exposed to PAW showed a steady downward trend in general, presenting a value of 228 U/min at 5 d.

RS can inactivate enzymes by preventing the substrate and coenzyme binding and subsequent catalysis, mainly due to conformational changes around the active site and chemical modification of the active site residues [25]. It can be inferred that ROS and RNS can be dissolved in water to continuously inactivate PPO, as shown in the PAW group. However, these active substances may directly react with phenolic substances on the mushroom surface to form quinones to accelerate browning, accounting for the significant browning in the DBD group (Figure 2B). Surowsky et al. [32] and Pankaj et al. [33] hypothesized that OH, O_2_^−^, HOO, and NO radicals modified the chemically reactive side-chains of amino acids, such as cysteine and the aromatic rings of phenylalanine, tyrosine, and tryptophan, consequently leading to a loss in enzymatic activity. It is speculated that the reactive radical species produced after plasma-activated water treatment damage the PPO structure, further decreasing its activity, which is irreversible.

## 4. Conclusions

The visual browning, BI, and *∆E* of the mushrooms increase gradually during 5 d of storage at +4 °C. The PAW sample displays the lowest *ΔE* values and cap BIs for both sides. Contrarily, direct ACP treatment increases the overall color variation in the mushrooms during storage. The PAW group exhibits maximum tissue firmness and a significantly (*p* < 0.05) lower weight loss. The sensory score results indicate that the plasma-activated water-treated button mushrooms may be the most popular with consumers. Furthermore, the plasma-activated water treatment may irreversibly affect the PPO activity. Further research is expected to reveal the types of RS produced by ACP and their influence on the PPO structure. In summary, plasma-activated water is most successful in preserving button mushrooms and is expected to become a new green preservation method.

## Figures and Tables

**Figure 1 foods-11-03504-f001:**
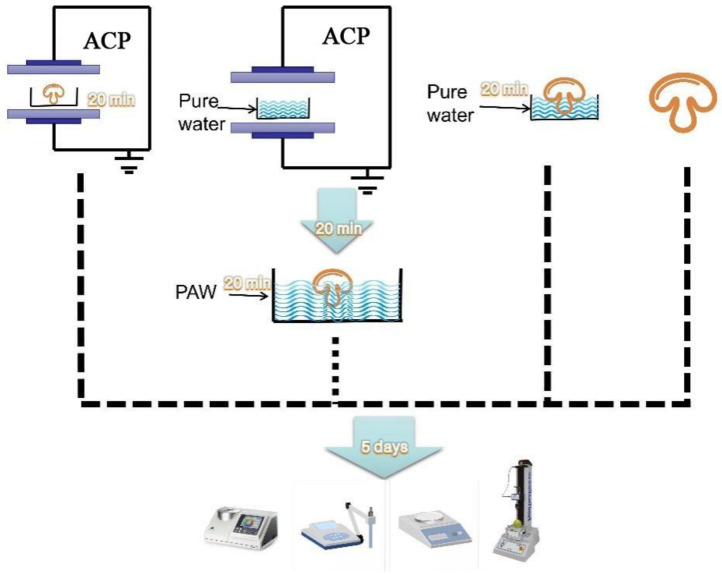
The study was divided into four groups, from left to right: ACP generation by dielectric barrier discharge plasma equipment to directly treat button mushrooms (DBD); Plasma-activated water generation by dielectric barrier discharge plasma equipment to soak button mushrooms (PAW); Pure water directly soak button mushrooms (PW); untreated button mushrooms (control).

**Figure 2 foods-11-03504-f002:**
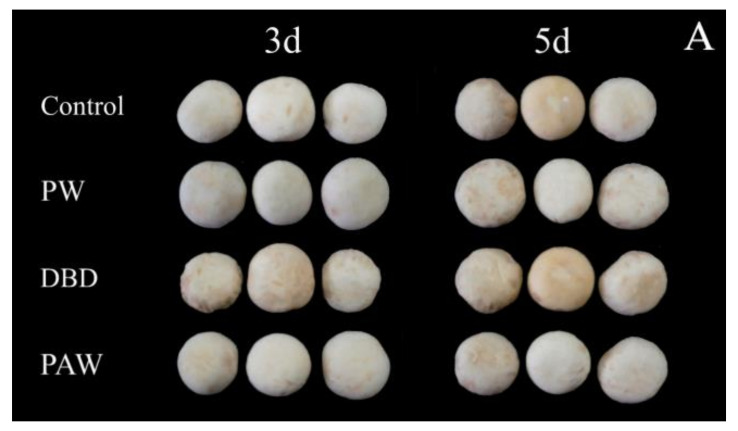
The effect of different treatments on the (**A**) visual quality, (**B**) BI, and (**C**) ∆*E* of the button mushrooms. Each value is expressed as a mean ± SD. Different letters indicate significant differences (*p* < 0.05).

**Figure 3 foods-11-03504-f003:**
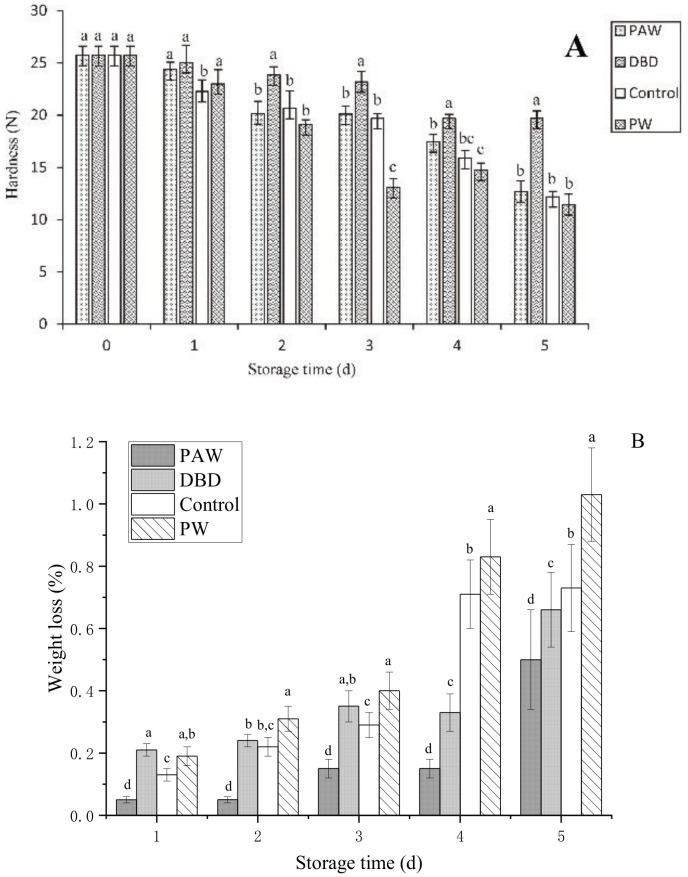
(**A**) The firmness and (**B**) weight loss of the PAW-, DBD-, PW-treated, and control button mushrooms during storage. ANOVA results are represented by letter notation in the bar graphs of panels (**A**,**B**), where groups marked with different letters differ significantly (*p* < 0.05).

**Figure 4 foods-11-03504-f004:**
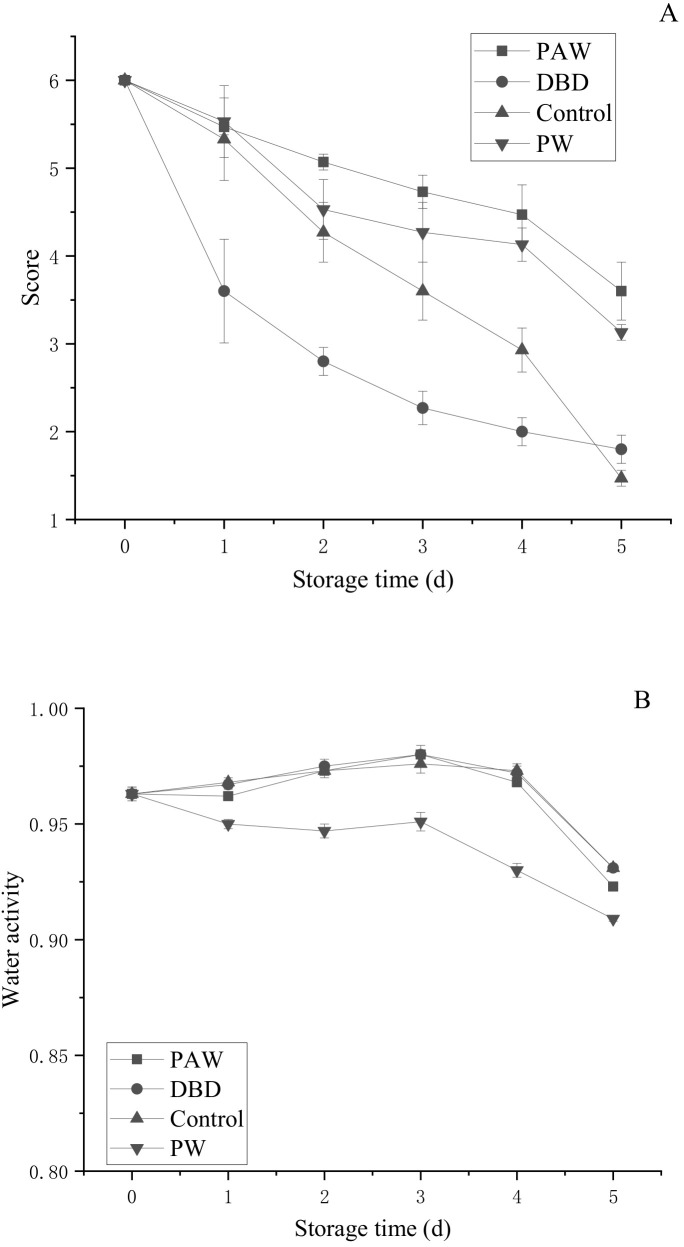
Changes in the (**A**) sensory evaluation and (**B**) water activity of the button mushroom in different treatment groups.

**Figure 5 foods-11-03504-f005:**
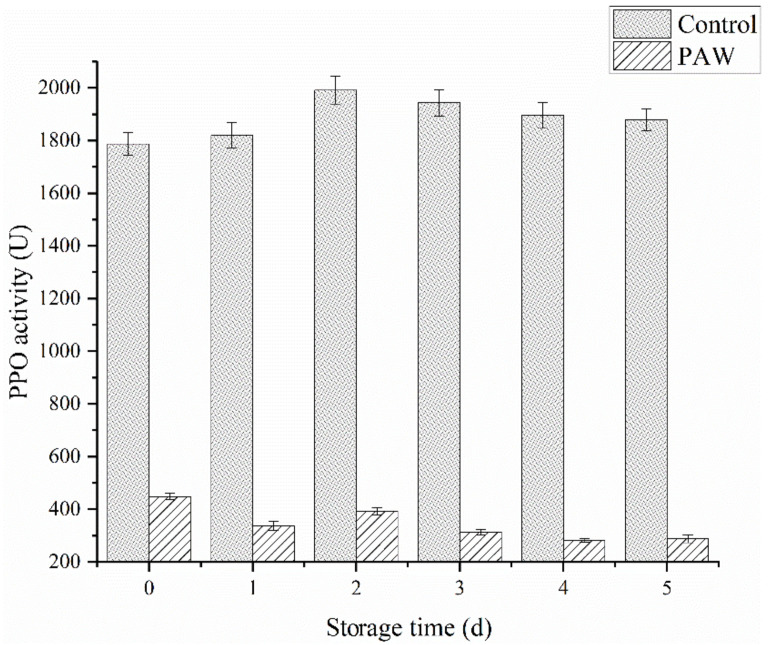
The PPO activity of the control and PAW groups during storage.

**Table 1 foods-11-03504-t001:** The sensory evaluation standards for button mushrooms during storage.

Level	Color	Umbrella Opening Degree (%)	Wilting Degree (%)	Score
1	White	<3	<5	6
2	Slight color change	3–10	5–10	5
3	Color deepening	10–20	10–25	4
4	Slight browning	20–50	25–50	3
5	Brown	50–80	50–80	2
6	Dark brown	>80	>80	1

**Table 2 foods-11-03504-t002:** The pH values of the PAW-, DBD-, PW-treated, and control button mushroom samples during storage.

Day	DBD	PAW	PW	Control
0	6.14 ± 0.04	6.14 ± 0.02	6.14 ± 0.05	6.14 ± 0.02
1	6.07 ± 0.02	6.02 ± 0.01	6.06 ± 0.03	6.08 ± 0.06
2	6.02 ± 0.06	6.01 ± 0.01	5.95 ± 0.02	5.98 ± 0.04
3	6.04 ± 0.04	6.01 ± 0.04	6.06 ± 0.01	6.07 ± 0.04
4	6.14 ± 0.01	6.08 ± 0.02	6.02 ± 0.04	5.99 ± 0.03
5	6.18 ± 0.02	6.08 ± 0.02	6.14 ± 0.03	6.11 ± 0.01

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
