# Peer review of "The Effect of Dielectric Barrier Discharge Plasma Gas and Plasma-Activated Water on the Physicochemical Changes in Button Mushrooms (Agaricus bisporus)"

_foods, 2022, doi:10.3390/foods11213504_

Round 1

Reviewer 1 Report

Dear authors, 

great work You have done...

please check some minor suggestion changes

Check the missing of dots.

Author Response

Great work you have done...

Answer: Thank you very much for the positive comments.

please check some minor suggestion changes

Answer: Thank you for suggesting these changes to our manuscript, which make my article even better. I have worked on revisions for each suggestion.

Check the missing of dots.

Answer: Thank you again for your very good advice on my manuscript. I have checked the dots in the manuscript and made additions.

Reviewer 2 Report

General comment

The manuscript provides some interesting results about the use of plasma as a food perservation technology in mushrooms. However, as the authors mention in the 2.3 Sample Treatments section, the sample size is very small (30 replicates in each group whilst only 3 samples were used for the analysis each time). Hence, my main concern is that the presented results are very specific to the conducted experiments and the analysed sample. Also the sample size is very small to conduct statistical analysis.  

Specific comments:

- Bibliography should be enriched.  

- Line 155: Please add the pH and the concentration of the phosphate buffer used 

- Line 157: Same as above for the sodium phosphate buffer

- The PPO activity was not monitored in mushrooms in storage but in the stored PPO extract. Therefore the observed decrease in PPO activity might not be due to the plasma treatment but to a possible loss in the enzyme activity in the extract during storage. 

Reviewer 3 Report

The manuscript entitled is interesting and innovative. It describes the effects on quality of button mushrooms after the application of atmospheric cold plasma to extend its shelf life.

The materials and methods section are well described, and the results are discussed and analysed appropriately. Conclusions are supported by the results. The figures, tables and the references are adequate.

I would suggest changing Fig.4 by a Table because there are not notably changes in pH value.

Author Response

The manuscript entitled is interesting and innovative. It describes the effects on quality of button mushrooms after the application of atmospheric cold plasma to extend its shelf life. The materials and methods section are well described, and the results are discussed and analysed appropriately. Conclusions are supported by the results. The figures, tables and the references are adequate.

Answer: Thank you very much for the positive comments.

I would suggest changing Fig.4 by a Table because there are not notably changes in pH value.

Answer: We removed Figure 4 and turned it into a table.

Reviewer 4 Report

1. The introduction focuses on the nutraceutical properties of button mushrooms, and these have not been determined in the experiment. Please focus the introduction on the parameters evaluated in the research.

2. The introduction does not make clear the novelty of this research. Please include it.

3. Include the quality parameters that were evaluated to select the raw material.

4. Section 2.2. Please indicate what is the room temperature.

5. The information in table 2 can be incorporated into the text. Please remove table 2.

6. Figure 1. Please be more descriptive in the figure caption.

7. Figure 3 A. Please include statistic data.

8. Figure 4. Can be deleted and mention in the text that there are no significant differences.

9. Why were the changes in the main bioactive compounds present in the button mushrooms not measured?

10. Why were the microbiological parameters not evaluated?

Author Response

  1. The introduction focuses on the nutraceutical properties of button mushrooms, and these have not been determined in the experiment. Please focus the introduction on the parameters evaluated in the research.

Answer: We have modified the introduction and added the introduction of parameters related to this article.

  1. The introduction does not make clear the novelty of this research. Please include it.

Answer: We have revised the last paragraph of the introduction to indicate the innovation of this article. “In this study, we introduced the ACP preservation technology into the study of button mushroom freshness change during 4℃ storage. Using dielectric barrier discharge plasma equipment to generate ACP to directly treat button mushrooms (DBD) and dielectric barrier discharge plasma equipment to generate plasma-activated water to soak button mushroom (PAW) immersion to examine the physicochemical changes during storage. The results are compared to two additional groups of mushrooms, one treated with pure water (PW) and untreated button mushrooms (Control). Postharvest preservation provides new insight and a theoretical basis for conserving fresh food.”

  1. Include the quality parameters that were evaluated to select the raw material.

Answer: The supplementary explanation was made in chapter 2.4, the content is as follows: “Before the experiment, the original pH value, hardness, water activity and sensory score of each group of samples were measured, and there was no significant difference between the samples of different groups (p > 0.05).”

  1. Section 2.2. Please indicate what is the room temperature.

Answer: We checked the experimental records, and the room temperature was 25℃.

  1. The information in table 2 can be incorporated into the text. Please remove table 2.

Answer: We deleted Table 2.

  1. Figure 1. Please be more descriptive in the figure caption.

Answer: We modified the figure caption of Figure 1 and gave a detailed description of Figure 1.

  1. Figure 3 A. Please include statistic data.

Answer: We replotted Figure 3 A

  1. Figure 4. Can be deleted and mention in the text that there are no significant differences.

Answer: We removed Figure 4 and turned it into a table.

  1. Why were the changes in the main bioactive compounds present in the button mushrooms not measured?

Answer: Polyphenol oxidase (PPO) activity is a major factor affecting the degree of browning of mushrooms. In this study, we used catechol as the substrate for the enzymatic reaction. The treated mushrooms were lyophilized as powders, dispersed in phosphate buffer (0.05M, pH7.4), and enzymatically reacted with catechol dissolved in sodium phosphate buffer (0.05M, pH7.4) to calculate the activity of polyphenol oxidase in mushrooms. In addition, the main focus of this study is the change in storage quality, so the main oxidase activity index was measured here.

  1. Why were the microbiological parameters not evaluated?

Answer: High water content and a neutral pH in button mushrooms provide an ideal medium for microbial growth. But in our study, the button mushrooms were stored at 4℃ throughout the experiment. This temperature is not suitable for the growth and reproduction of microorganisms. Therefore, this study focused more on the effects of atmospheric cold plasma and plasma-activated water on the physicochemical properties and polyphenol oxidase activities of button mushrooms.

Round 2

Reviewer 2 Report

The manuscript has improved considerably after the last revision. However, I am afraid that my comments regarding the sample size and determination of PPO activity during storage have not been addressed appropriately; three replicates are very few, and the activity of PPO should have been determined in stored mushrooms not in the stored extract.

Author Response

First of all, thank you very sincerely for the positive comments. It is really regrettable that due to objective reasons such as limited funding, our experiment does not have a large enough sample size, but three parallel replicates are statistically significant. For example, in the document "Effect of Citric Acid on Relative Activity and Kinetic and Browning Parameters of Polyphenol Oxidase", three parallel repetitions are also used to illustrate the enzyme's activity.

Secondly, by consulting the literature, in the literature referenced in this article, “Differential in vitro inhibition of polyphenoloxidase from a wild edible mushroom Lactarius salmonicolor”, PPO activity in mushroom samples was determined also using a method “enzyme kinetics and substrate specificity”. This method is to monitor the change in the concentration of a given substrate during the enzymatic reaction and carries out the mirror by the OD value. Corresponding to this study, catechol was added to frozen pulverized mushroom samples as a substrate, and the changes in enzymatic activity during the process were measured at different time points. A reference was also added to further illustrate the ability to use catechol as a substrate to measure the PPO enzymatic activity of a sample.

Finally, thank you again for your valuable comments on the manuscript, which are helpful for the improvement of our future research.

Reviewer 4 Report

.

Author Response

Thank you for reviewing this study. I have worked on revisions for each suggestion. Thanks again for your hard work.Best wishes to you.